# Lung Nodule Segmentation with a Region-Based Fast Marching Method

**DOI:** 10.3390/s21051908

**Published:** 2021-03-09

**Authors:** Marko Savic, Yanhe Ma, Giovanni Ramponi, Weiwei Du, Yahui Peng

**Affiliations:** 1Department of Engineering and Architecture, University of Trieste, Piazzale Europa 1, 34127 Trieste, Italy; marko.savic@studenti.units.it; 2Information and Human Science, Kyoto Institute of Technology, Hachigami-cho, Sakyo-ku, Kyoto 6068585, Japan; duweiwei@kit.ac.jp; 3Tianjin Chest Hospital, Tianjin 300051, China; 5020200630@nankai.edu.cn; 4School of Electronic and Information Engineering, Beijing Jiaotong University, Beijing 100044, China; yhpeng@bjtu.edu.cn

**Keywords:** segmentation, fast marching method, lung nodules, computed tomography, lung phantom

## Abstract

When dealing with computed tomography volume data, the accurate segmentation of lung nodules is of great importance to lung cancer analysis and diagnosis, being a vital part of computer-aided diagnosis systems. However, due to the variety of lung nodules and the similarity of visual characteristics for nodules and their surroundings, robust segmentation of nodules becomes a challenging problem. A segmentation algorithm based on the fast marching method is proposed that separates the image into regions with similar features, which are then merged by combining regions growing with k-means. An evaluation was performed with two distinct methods (objective and subjective) that were applied on two different datasets, containing simulation data generated for this study and real patient data, respectively. The objective experimental results show that the proposed technique can accurately segment nodules, especially in solid cases, given the mean Dice scores of 0.933 and 0.901 for round and irregular nodules. For non-solid and cavitary nodules the performance dropped—0.799 and 0.614 mean Dice scores, respectively. The proposed method was compared to active contour models and to two modern deep learning networks. It reached better overall accuracy than active contour models, having comparable results to DBResNet but lesser accuracy than 3D-UNet. The results show promise for the proposed method in computer-aided diagnosis applications.

## 1. Introduction

### 1.1. General

Lung cancer is one of the most lethal cancers in the world, and its 5-year-survival rate is only 18% [1]. Noninvasive imaging is often used in clinical practice, and computed tomography (CT) images have a crucial role in early lung cancer diagnosis and survival time improvement [2]. The National Lung Screening Trial (NLST) showed a 20% decline in mortality specific to lung cancer, stressing the importance of nodule detection and assessment [3,4]. With the development of CT technology, images can be generated in mere seconds and a single scan contains at least a few hundred images that radiologists have to analyze. This can be very challenging work because radiologists must look at a large number of images and detect lung nodules without mistakes; as a consequence, the work can be a heavy physical and mental burden. Moreover, even if a lung nodule is detected, in some cases it is difficult to judge its benignity or malignancy, because there are not enough clear features to form an accurate classification. Computer-aided diagnosis (CADx) systems for automatic diagnosis of pulmonary diseases and lung cancer have been devised over the years to assist in this endeavor. These systems mainly depend on the segmentation of different pulmonary components. Computer-aided detection (CADe) is usually restricted to marking the visible parts or structures in an image, whereas CADx helps to evaluate the structures identified in CADe [5]. In this paper we provide a tool that can perform measurements on already detected nodules, belonging to the CADx domain. Automated measurements speed up screening activities and provide less dispersed results than subjective criteria by different radiologists. Its performance was evaluated using objective criteria such as Dice score and diameter precision. Nodule segmentation is fundamental for automated lung cancer diagnosis and the predictors most widely used to assess probability of malignancy are nodule size, shape and growth rate [6], all of which are dependent on the accuracy of the segmentation. The task of segmenting lung nodules poses many challenges and is plagued by their heterogeneity in CT images. Cases that are problematic to segmentation arise when the surrounding tissue shares similar visual characteristics with the nodules, making it hard for segmentation algorithms to draw a clear boundary between them. Instances where nodules adhere to the pleura or to blood vessels can be difficult, as the image intensity difference is often imperceptible. This problem emerges also in sub-solid nodules, which are characterized by low contrast to the surrounding tissue. Cavitary cases, which present a high-intensity differences for distinct parts of the nodule, are also arduous. Additionally, when the diameter is small, the nodule can be hard to distinguish from its surroundings and background noise. Furthermore, in case radiologists need to draw the nodule’s contours, the task becomes much less time consuming with a reliable tool that can draw an initial boundary. Lung phantoms are used to assess the accuracy of volumetric, low-dose CT at detecting changes in nodule volume [7,8] and to evaluate the performance of computer-aided detection [9]. In this paper, a lung nodule phantom is employed to evaluate segmentation accuracy.

### 1.2. Related Work

In view of the exceptional importance of lung disease detection and diagnosis, large efforts have been made over the years to improve multiple correlated fields of research, including lung nodule segmentation, lung segmentation, lung nodule detection and diagnosis of other lung diseases.

Regarding lung nodule segmentation, which is the main topic of this work, many advancements have been made and can be distinguished in traditional image processing-based and machine learning-based techniques.

Traditional methods include different approaches: morphological operators, region growing, energy optimization, fuzzy methods and other more specialized methods. Numerous methods based on morphological operators were proposed to address nodule segmentation. Kostis et al. [10] applied iterative morphological filtering to remove vessels affixed to solid nodules. However, even though generally these methods are fast and easy to implement, the size of the morphological operator is difficult to calibrate due to the variety of nodule dimensions. Kuhnigk et al. [11] carried out a more complex morphological correction which allows one to manage nodules regardless of size. In region growing methods, starting from a specified seed point, pixels are included into the segmentation iteratively if they satisfy a criterion. This approach can have difficulties converging in juxtapleural cases due to the similarity between the nodules and surrounding regions [12]. To counter this issue, Dehmeshki et al. [13] proposed a new contrast-based region growing method that employs a fuzzy connectivity map, although it did not perform well with irregular nodules. The core issue in region growing methods is to properly set a merging criterion, so that the segmentation is stable and accurate. In energy optimization methods, the segmentation problem is formulated within an energy minimization framework. Level set methods are a popular implementation of this idea, where images are represented as level sets that are evolved iteratively to minimize an energy [14]. To better handle juxtapleural nodules, Farag et al. [15] used the shape prior hypothesis along with level sets. Boykov and Kolmogorov et al. [16] framed the problem within a maximum flow optimization framework and applied a graph cut method for segmentation. Energy optimization methods are generally not well suited to ground glass opacity (GGO) and juxtapleural nodules. There has been research aimed at improving performance with GGO nodules with the expectation–maximization algorithm [17] and active contour models (ACM) [18], where for part-solid nodules the solid and non-solid parts are segmented separately and combined. Several algorithms have been developed that exploit fuzzy C-means clustering, among which is the method proposed by Li et al. [19] that incorporates Gaussian mixture models’ prior knowledge into the traditional fuzzy C-means algorithm to improve robustness. The main drawbacks of fuzzy C-means segmentation methods are noise sensitivity, outliers and initial cluster selection. Various specialized techniques have emerged that tackle specific types of nodules. To quickly extract contours of juxtapleural nodules, Mekali et al. [20] proposed a detection and segmentation method based on lung boundary pixel extraction, concave point extraction and separation of pleural values from nodules. To deal with inhomogeneity and fuzzy contours, Wang et al. [21] used an approach which combined the enhanced total-variance pyramid and grab cut to segment solitary nodules. They incorporated multiscale information to optimize the edge term and extract the nodule boundary with a Gibbs energy functional. Despite their advantages, techniques that are specialized to only one type of nodule naturally lack flexibility and are difficult to apply in general settings.

In recent years, an increasing number of studies have developed artificial intelligence tools in the field of medical image segmentation. Lu et al. [22] have proposed a stratified learning framework including supervised image segmentation. Hu et al. [23] segmented the lungs’ area and carried out a Hessian-based vascular feature extraction procedure to remove the blood vessels and obtain candidate nodules, then classified them with the aid of a neural network. Gonçalves et al. [24] developed a different approach that uses Hessian-based strategies for lung nodule segmentation—a multiscale process that uses the central medialness adaptive principle. The method proposed by Jung et al. [25] can separate solid and non-solid components in GGO nodules using an asymmetric, multi-phase, deformable model. Due to their increased efficiency in terms of computation and number of parameters, deeper networks have obtained great success and deep learning has become ubiquitous in most medical image processing applications, including lung nodule segmentation. A convolutional neural network (CNN) is a multi-layered neural network well adapted to image processing which learns hierarchical representations of an image at every layer, with the last layer usually corresponding to a label. Thus, segmentation turns into a voxel classification problem. For instance, Wang et al. [26] introduced the multi-view convolutional neural network (MVCNN) for nodule segmentation, which focused on extracting a miscellaneous set of features from axial, coronal and sagittal views. Ronneberger et al. [27] proposed 2D U-Net architecture with an approach which was well suited to biomedical imaging. However, due to the fact that CT images are originally three-dimensional, the down-sampling and subsequent up-sampling causes spatial information to be lost. Variations of CNN, such as the central focused convolutional neural network (CF-CNN) proposed by Wang et al. [28], and the dual-branch residual network (DBResNet) for lung nodule segmentation proposed by Cao et al. [29], have achieved competitive performance. By targeting pure GGO nodules specifically, Qi et al. [30] performed segmentation on initial and follow-up CT scans using a CAD system based on CCN, and subsequently measuring and analyzing the nodules to determine growth and risk factors. In most recent works the U-Net architecture has been popular with three-dimensional implementations that improve its performance. Funke et al. [31] trained a 3D-UNet model using using the STAPLE algorithm, and Xiao et al. [32] combined the 3D-UNet and Res2Net architectures to create a new model which reached a Dice score of 95.3% on the Lung Image Database Consortium (LIDC) dataset. In a different approach, Hu et al. [33] utilized a hybrid attention mechanism and densely connected convolutional networks, reaching a Dice score of 94.6%.

Despite the remarkable advances in performance attributed to deep learning methods, there are still several disadvantages which are to be considered. Deep learning requires large amounts of data, which can be especially laborious to prepare for supervised tasks where data labeled by experts are necessary, such as in CT scans and X-rays. This issue has been tackled by applying data augmentation techniques. The most widely used are translations, rotations, resizing, flipping and cropping patches [34]. There have also been more sophisticated approaches. For example, Gao et al. [35] utilized a generative adversarial network to create more training data to improve lung nodule detection. Another problem related to the data is class imbalance, which leads the network to specialize only in specific classes that have the most samples. Simple solutions to this problem are oversampling, undersampling and data augmentation. Overfitting is another issue, which causes the networks to overly specialize on the training data, and it has been reduced in many ways, such as data augmentation, cross-validation, regularization, batch normalization and pooling. A more practical drawback of deep learning is the need for powerful computers. Indeed, there is a tradeoff between the performance and complexity of the network, which encourages the use of more powerful machines to train and execute the network.

Apart from lung nodule segmentation, there are other closely related problems which have been studied in order to improve the diagnosis of pulmonary diseases.

One such task is lung segmentation. Most often it is a necessary step in preprocessing and has been tackled in different ways. Kavitha et al. [36] approached the problem with a novel strip method and a marker-watershed method based on particle swarm optimization (PSO) and fuzzy c-means clustering. Kim et al. [37] used a deep learning method based on the U-Net architecture, where self-attention is exploited to reach better performance in lung segmentation in chest X-rays.

Another relevant problem is lung nodule detection. Most often detection and segmentation are incorporated to produce end-to-end systems; such a method was proposed by Mekali et al. [20] to tackle juxtapleural nodules with a traditional approach. Recently, deep learning has been a predominant solution to this task. Huang et al. [38] created an end-to-end arrangement, combining detection performed via a regional-CNN and segmentation via a fully convolutional network (FCN).

Lung cancer is just one of the many pulmonary diseases which have been studied by methods that analyze CT and X-ray scans in order to detect abnormal tissue. To detect diseased lung tissue, Polap et al. [39] proposed lung segmentation combined with a bio-inspired algorithm, resulting in an automated assistance methodology which can help radiologists quickly identify unhealthy tissue. To address the detection of pulmonary diseases, various hybrid procedures have emerged which combine neural networks with heuristic or fuzzy methods. Ke et al. [40] used a neural network for a preselection phase: if the input X-ray image was determined to contain unhealthy tissue, it was passed on to the detection phase. The detection phase was handled by a heuristic search over the image which ascertained which pixels belonged to degenerated tissue. This neuro-heuristic method reached about 79% accuracy in detection and has shown that neural networks are efficient in the preselection phase. Adaptive neuro-fuzzy inference systems (ANFIS), which integrate both neural networks and fuzzy logic principles, have also been applied to similar tasks. For example, Santoso et al. [41] used this method for the detection of pneumonia and pulmonary tuberculosis. To undertake diagnosis of COVID-19, Ukaoha et al. [42] employed ANFIS and reached an accuracy of 96.6%. A different novel solution for COVID-19 diagnosis in CT scans was proposed by Akram et al. [43]. The technique consists of feature extraction followed by a selector based on an optimized genetic algorithm. Finally, the selected features are used as input for a naive Bayes classifier, which resulted in a 92.6% acccuracy. A deep convolutional approach was taken by Wang et al. [44] to build a network architecture tailored for COVID-19 detection in chest X-rays. The network reached an accuracy of 99.1% with a lightweight design.

All the related works mentioned and discussed are summarized in Table 1.

### 1.3. Objective

This study proposes an algorithm based on the fast marching method (FMM) that can accurately segment lung nodules in CT images, creating a powerful tool that can be employed in CADx systems. This approach can help radiologists quickly and accurately find the contours of lung nodules without needing powerful computers and complex learning models. The proposed technique is based upon existing work done in the field of image processing, which has been combined in a novel way and applied to the problem of lung nodule segmentation. Regarding the same problem, this study means to overcome several limitations of existing methods. One typical weakness is the sensitivity of merging criteria in region growing methods, which was tackled in order to provide more stable convergence. Another limitation is over-specialization of methods. As a matter of fact, the technique proposed in this work aims to accurately segment different nodule types with varied surroundings. Indeed, the heterogeneity of nodules was taken into account at the development stage and the evaluation was addressed categorically. This method segments nodules in a simple and fast way, in contrast to deep learning techniques, which require much higher computational costs to yield quality improvements. The performance will be quantified by experimenting with a hybrid method of segmentation evaluation, by combining results from an objective method employed on the simulation data and a subjective method employed on real patient data. During evaluation, the simulated lung nodules will be monitored to check whether they successfully emulate features of real lung nodules.

### 1.4. A Summary of the Novel Contributions

This work was based and built upon existing image processing techniques, with the following original contributions:Better handling of juxtapleural nodules by refining segmentation of the lungs’ area during the preprocessing stage, by using convex hulls instead of traditional morphological operators, and by devising a padding scheme that counters the undesirable effect the pleura has on the speed function.The region-based method was inspired by existing multi-label fast marching method implementations, with the original addition of using k-means to cluster the regions before merging, resulting in more robust segmentation without having to sacrifice detail. An existing automatic seed grid generation method has been improved, making it more suitable for lung nodule segmentation, by utilizing the gradient mean as the threshold and shifting seeds diagonally as well. Furthermore, the method for selecting the first cluster to be merged is original, and it combines distance from the center and mean intensity to make the decision.Evaluating the performance with both an objective method and a subjective method for comparison—in this way the evaluation can be done in greater depth, recognizing possible biases each method can have. For the objective method a simulated nodule dataset was created from lung phantom scans. For the subjective method a questionnaire was designed with which each segmentation instance was rated by subjects experienced in radiology.

## 2. Methods

The segmentation technique is based on the fast marching method [45] with multiple seed points to produce a set of regions in the image that will be merged accordingly in order to segment the nodule. This method is a consistent, accurate and highly efficient algorithm based on fully utilizing entropy-satisfying upwind schemes and fast sorting techniques. It was chosen as the core of the segmentation algorithm due to its efficiency: the algorithm sweeps through a grid of *N* points in Nlog(N) steps to obtain the evolving time position of the front as it propagates through the grid. Several applications of the fast marching method in image segmentation were proposed; a multi-label version of the algorithm was introduced [46], as was a generalized method [47]. Specific applications in medical image processing have yielded successful results; in [48], FMM was used for automatic liver segmentation in CT images. The variation of the fast marching method that will be discussed is specifically tailored for lung nodule segmentation and consists of ideas and methods inspired by previous work that have been combined in a novel way.

The segmentation method is divided into three parts:Input preparation: By utilizing the nodule image, a grid of starting points (or seeds) and a speed function are created. Both these elements are necessary for executing FMM. The seed grid generation method was drawn from [49]; it was reshaped the concept to better fit lung nodule segmentation (see Section 2.2.1).Fast marching method: It is applied iteratively until an arrival time is calculated for every pixel and every pixel is assigned to a region. The end results are two matrices, the first containing all the arrival times and the second containing labeled regions. The base implementation was performed by following the algorithm as described by J.A. Sethian in [45], with the addition of a region assigning component [46] (see Section 2.2.2).Region merging: The regions are merged into a final segmentation mask by utilizing a region growing method aided by the k-means algorithm. This new way of combining FMM with k-means greatly improves robustness, without having to sacrifice detail (see Section 2.2.3).

### 2.1. Background

The fast marching method [45] is an algorithm that was developed as a quick numerical scheme for solving the Eikonal equation, which is a non-linear partial differential equation found in wave propagation. It is utilized in many diverse applications, including path planning, optimal control, computational geometry, computational fluid dynamics and image processing. Its formulation is shown in Equation (Equation 1).
(1)$$\left\{ \begin{array}{cc} \left|\nabla u(x)|f(x\right) & =1\;\forall x\in \Omega \subset R^{n}\\ u(x) & =q(x)\;\forall x\in \partial \Omega \end{array}\right.$$
where $f\left(x\right):\bar{\Omega }\to \left(0,+\infty \right)$ is a positive speed function, q(x):∂Ω→[0,+∞) is the exit time-penalty and u(x) is the value function, i.e., the minimum time to exit $\bar{\Omega }$ through ∂Ω starting from a point x∈Ω. This interpretation for the solution of Equation (Equation 1) comes from isotropic time-optimal control problems [50]. In practice, firstly the speed function is defined for every point in the spatial grid and a starting point is chosen, from which arrival times are calculated iteratively using the fact that information only flows outward. This problem is a special case of level set methods [51], namely, it is a more efficient and discrete application. While taking a two dimensional case with a front moving with speed F(x,y)>0, let us suppose that for the propagating curve we graph the evolving zero level set above the xy plane. That is, let T(x,y) be the time at which the curve crosses the point (x,y). Then the surface T(x,y) satisfies Equation (Equation 2).
(2)|∇T|F=1

The gradient of the arrival time surface is inversely proportional to the speed of the front. The gradient of T is approximated as seen in [52] and each arrival time is calculated with the method found in [53]. Having a uniformly sized spatial grid, the points (x,y) can be denoted using two indices (i,j). The main idea underlying the fast marching method is to systematically construct the solution in a downwind fashion, stemming from the fact that the upwind difference structure of the arrival time equations means that the information propagates outward, from smaller values of $T_{ij}$ to larger values [54]. Only a small subset of points around the front, called the narrow band, are examined at each iteration, making the algorithm fast. The aim is to propagate the front by updating the narrow band in an efficient manner [55].

The technique is most easily explained algorithmically [45]. Considering a grid of equidistant points and a set of seed points, for each point a speed $F_{ij}$ is defined and the objective is to calculate each arrival time $T_{ij}$. At first all the points are set to their initial values, and then the algorithm marches forward in a loop.

Setting Initial Values:1.(Known points, K): The seed points are added to set K; this will be the initial front. The most simple case is just one seed point; for multiple seeds the procedure is similar. Let S be the set of seed points; set $T_{ij}=0$∀(i,j) ∈S2.(Near points, narrow band,N): Let N be the set of all grid points that are neighboring the known points; in the simplest case it is just the four neighbors of the single seed point.Set $T_{ij}=\frac{1}{F_{ij}}$∀(i,j) ∈N, which are the first calculated arrival times.3.(Far away points,F): Let far away points be defined as all the points in the grid that are not known or near. Set $T_{ij}=\infty$∀(i,j) ∈F.

Marching Forward (as shown in Algorithm 1):
**Algorithm 1:** FMM forward marching loop
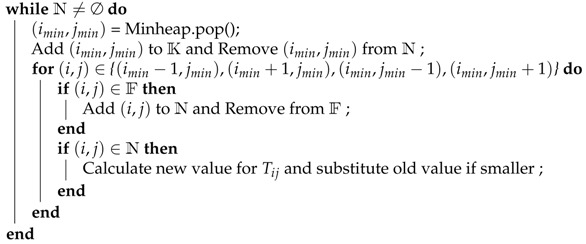


This technique is made efficient by storing the values of T for each point belonging to the narrow band in a minimum heap structure. The smallest time value can be instantaneously obtained by popping the root element of the heap.

### 2.2. Proposed Method

To separate the image in regions using the FMM algorithm, it is necessary to create multiple propagating fronts starting from multiple seed points. For each seed, a region is defined which at first only contains the seed and then expands to all the points that the seed reaches in the shortest time. This functionality can be added to the algorithm by taking note whenever a new time is calculated, for when the seed propagated. When the algorithm is finished and all points are known, every point has been assigned to a region, creating sort of a “stained glass” effect. In Figure 1 an example is shown; in this case there are two seeds that propagate; the darker colors symbolize the known points (final values) and the brighter colors the narrow band (trial values); the arrows show which points in the narrow band have been just updated. The speed function is very simple; since the values on the right side are higher, seed B will expand into a larger region by reaching the middle points in less time. The initial order in which the seeds are set to known is irrelevant.

The proposed method can be divided into three stages, as can be seen in Figure 2. The first stage generates a seed grid and a speed function which will be the inputs of the next stage, which will implement the region-based multiple-front FMM algorithm, resulting in a set of regions. To obtain the segmentation mask, the regions are firstly clustered using k-means and then recursively merged depending on a mean intensity criteria. The different stages will be explained thoroughly in the following subsections.

#### 2.2.1. Input Preparation

The speed function depends on the gradient of the input, and the seeds are distributed on a grid based on the local gradient mean. Since the speed function is an image with the same dimensions as the input, it can also be referred to as speed image. A good propagation speed image for segmentation is close to zero near object boundaries and relatively high in between. To obtain the speed image, at first the input image is denoised with an anisotropic diffusion filter, which removes noise without influencing edges. Afterwards, the magnitude of the image gradient is computed. The speed image needs to be an inverse function of the image gradient; the function chosen was an exponential with a negative exponent—specifically, $F\left(x,y\right)=e^{-2\nabla (x,y)}$. After normalizing the speed image, the values are near zero at the boundaries, and in homogeneous regions they are close to one. In the case of jutxapleural nodules, an issue arises: since the pleura are removed during the lung segmentation step in the preprocessing section, there will be a high image gradient because of the sharp transition, and this will influence the segmentation negatively. To render the image gradient, and subsequently the speed image, invariant to this undesired effect, a padding scheme was devised and applied to the input before calculating the gradient.

An optimal seed grid should have a higher concentration of seeds in uniform areas in the image; a good way to satisfy this condition is to use the normalized image gradient as a criterion. Another thing to consider is that the images that need to be segmented are low resolution; this means that in order to obtain accurate results, a dense seed grid is needed. Keeping these two requirements in mind, the algorithm works in the following way:At first the mean of the image gradient is calculated and will be used as a decision threshold; this variable is called total grad mean.The seed grid is initialized with equidistant points; the distance can be varied, but to get the densest possible grid, the minimum value is chosen, which is three. For each seed a local area is assigned; with distance equal to three, the local area corresponds to the 8-connected neighborhood. For each seed’s local area, a mean is calculated using the gradient values; this variable is called local grad mean(i), with i being an integer that identifies the seeds.Based on the value of local grad mean(i) for seed i, three possible things can happen:1.If local grad mean(i)>β·total grad mean, then seed *i* is deleted from the grid. The local area is not sufficiently uniform.2.If α·total grad mean≤local grad mean(i)≤β·total grad mean, then seed *i* is shifted. For each point in the local area, the local gradient mean is calculated; the point corresponding to the minimum value is chosen as seed *i*, thereby moving it towards a more uniform area.3.If α·localgradmean(i)<totalgradmean, then seed *i* is kept as is; the local area is sufficiently uniform.

The parameters that define the thresholds for keeping/shifting/deleting seeds have been fixed to the values of α=1 and β=2. These values have been selected after extensive experimentation.

An example is shown in Figure 3; since the nodule is juxtapleural, the outside of the lungs is ignored in the grid creation, and the addition of shifting renders the grid denser and better distributed.

#### 2.2.2. Fast Marching Method

The speed matrix *F* and the seed points will be the inputs; the algorithm is run iteratively by using the techniques previously discussed, until all points are known. Finally, the end result will be two matrices with the same size as *F*:*T*: the times matrix containing all the shortest arrival times.*R*: the regions matrix carrying the information of which point belongs to which region.

Examples are shown in Figure 4 and Figure 5, where i is the iteration counter.

#### 2.2.3. Region Merging

The segmentation mask is obtained via a region growing method, starting from a region that is sure to belong to the nodule neighboring regions are added recursively if they match a certain criteria. The selected criteria is the mean intensity of each region, if the mean intensity difference between two regions is higher than the threshold this means that there should be a boundary between them, conversely if it is lower than the threshold they should both belong to the nodule. Since the regions are relatively small, due to the densely packed seeds, neighboring regions will often have a mean difference that is lower than the threshold even in presence of a boundary. A way to remedy the aforementioned issue, without sacrificing seed density, would be to group together regions into larger ones. Looking at the way the seed grid is constructed, seeds in uniform areas are packed closer together and should be grouped together based on Euclidean distance. This step was implemented by using k-means to put regions together into clusters. The default choice for initial centroids for the k-means algorithm is random, but this has the disadvantage of giving non-repeatable and varied results, often obtaining clustering that worsens segmentation performance. It makes sense to apply the same method used for the seed grid generation, because optimally the cluster centroids should stay away from the nodule border. For picking the initial centroids the distance can be varied to improve segmentation quality and this tunable parameter will be called cluster density. An example of how regions are grouped into clusters is shown in Figure 6.

The regions are bundled into clusters with a more pronounced mean intensity difference between them, consequently the merging criteria is more effective. A starting cluster needs to be chosen that is sure to belong to the nodule. When there are no obstructions or other tissue with similar radiodensity, a nodule is usually identified by a higher intensity in the image, but this is not always the case. The images that are used as input are centered on the nodules, but in edge cases like cavitary nodules the cluster containing the central pixel is not part of the nodule. Both closeness to the center of the image and high mean intensity are taken into account when selecting the starting cluster. For each cluster a ratio is calculated between the mean intensity and distance of centroid to the image centre, the cluster with the highest ratio wins and is selected as the starting point for merging. In Figure 7 an instance of cluster merging is shown.

#### 2.2.4. Parameter Tuning

There are several parameters which have set fixed and altering them does not change segmentation accuracy in a significant way. All the values were selected empirically. The fixed parameters are the following:σ=1: Before calculating the gradient magnitude the image is convolved with a Gaussian kernel with this standard deviation value.τ=−2: The speed function is an exponential of the gradient magnitude, the exponent τ is negative because it needs to be an inverse function of the gradient magnitude.α=1 and β=2: When generating the seed grid, these parameters determine the thresholds at which seed points will be kept, removed or shifted.*seed distance = 3*: This parameter determines how dense the seed grid is; the smallest possible value is picked because it provides the most detailed segmentation results. The side-effects of choosing this value are mediated by the addition of k-means clustering.

Furthermore, there are two tunable parameters that can be altered to improve segmentation:Cluster density: It determines how dense the cluster centroid grid is, thereby determining the size and total number of clusters. Lowering this parameter can improve segmentation accuracy, but the tradeoff is that the mean difference between neighboring clusters is less pronounced and thus an unwanted cluster (does not contain nodule tissue) could be merged. The default value of this parameter is 7.Mean threshold: It determines which clusters are suitable for merging and will be part of the segmentation mask. If the mean intensity difference between a cluster and its neighbor are below the threshold they are merged. The default value of this parameter is 0.15.

The two parameters have been set by exploring the space of possible values deemed effective, separately from each other, in multiple iterations in order to verify the inter-relationships of their effects. The criteria of choice were to maximise the perceived visual quality of the contour, and the spatial overlap between nodule and segmentation mask. Cluster density largely affected the perceived visual quality, while mean threshold mostly influenced the spatial overlap and stability of the segmentation. Future investigations will be necessary to study optimal setting of both parameters and to determine an automated procedure which maximizes the criteria.

## 3. Data

For this study two different datasets were used:Altered phantom dataset: It is comprised of nodules taken from lung phantom scans and modified using image processing tools. The CT images employed for this dataset have been specifically generated for this study by Tianjin chest hospital; the data were produced by scanning a lung phantom containing spherical nodules with different sizes.LIDC dataset: It is comprised of nodules selected from CT images taken from the Lung Image Database Consortium (LIDC) database [56]. Those scans were acquired from multi-detector row CT scanners with a wide range of scanning parameters [57].

The values or ranges from the two datasets are shown in Table 2, for the phantom scans the only parameter that changes between scans is the tube current (higher tube currents result in better image quality and reduced noise). Conversely the LIDC dataset contains a great number of different scans and the parameter range is broad, providing good variety.

The segmentation algorithm has been tested on both datasets and evaluated using two separate methods. This novel approach was taken for two reasons, the first being that this way the results of both evaluations can be compared in order to gain insight on which features the evaluation methods prioritize. The second reason is validation of the altered phantom dataset, by comparing the results it can be verified that the altered phantom nodules are in fact comparable to real patient nodules.

### 3.1. Altered Phantom Dataset

This dataset is entirely comprised of simulated lung nodules, it was constructed by modifying the CT data from the lung phantom scans. The advantage of using a phantom is having a more controlled and homogeneous environment to test the segmentation on, and compared to real patient data the simulated nodules have a predetermined size and shape. This way the ground truth images can be easily obtained, and there is no need for radiologist annotations. Spherical simulated nodules of three different sizes (5 mm, 10 mm and 20 mm) have been placed in a CT scanner and four different scans were performed, with varying tube currents. Higher tube currents give improved image quality and decreased image noise, but raise the radiation dose, which is not an issue when dealing with a phantom. The scans were performed with tube currents of 30 mA, 60 mA, 99 mA and 197 mA. The rest of the parameters were the same for every scan and can be seen in Table 2. A Philips Brilliance 256 iCT scanner was used for scanning, with a collimator width of 128 × 0.625 mm and rotational speed of the X-ray tube of 0.27 s/round. Idose3 was used for iterative reconstruction.

The lung phantom can be seen together with an axial slice in Figure 8.

The nodules found in this type of scans are of a circular shape and thus only represent a small portion of lung nodules. To overcome this lack of complexity and variety, the images were modified by using image processing tools. More specifically, the desired nodule boundary was selected and fitted to a new boundary using image warping and, in order to make the edges appear more realistic, the border was smudged. The ground truth image was also modified to fit the new boundary. These alterations were made so that the new simulated nodules could share similar characteristics with real nodules. Subsequently, a dataset containing 108 nodules of different sizes, shapes, surroundings and tube currents was created. The nodules were then categorized in an analogous way as the ones from the LIDC subset (see Section 3.2). The resulting categories were of varying size as is shown in Table 3. In addition, juxtapleural, juxtavascular, sub-solid and cavitary cases required extra steps to simulate their real features.

### 3.2. LIDC Dataset

The complete LIDC database contains 1018 scans, encompassing a large number of nodules with analogous features. A smaller representative subset (82 cases) was hand picked in order to have a varied set of nodules with different sizes and characteristics, making the subjective evaluation more manageable and efficient. Lung nodules can be classified in a number of different ways [58] based on their appearance and clinical significance. For sorting the nodules in different categories, a simple classification was devised based on visual appearance, by following guidelines from the Fleischner Society [59] and by taking a cue from the review presented by Shaukat et al. [60], with the addition of cavitary nodules taken from the glossary of terms for chest imaging proposed by the Fleischner Society [61]. Solid nodules were split in two categories to account for the vastly different boundaries they can have, and subcategories have been added to divide nodules based on their surroundings. The representative subset has been compiled and sorted in the following categories and subcategories, as shown in the examples in Figure 9.

Categories:Solid-round: These nodules have a fairly uniform image intensity and are characterized by a simple convex shape. They are a kind of solid nodules, the most common type, distinguished by homogeneous soft tissue attenuation. Nodules with this smooth and almost round shape are more likely to be benign.Solid-irregular: These nodules are also typified by a uniform intensity and belong to solid nodules, but have a more irregular shape. Nodules with irregular, lobulated or spiculated borders are associated with a progressively higher probability of malignancy than those with a smooth border.Sub-solid: The pixels containing the nodule have a highly varied image intensity and are not uniform. These correspond to ground glass opacifications (GGO) and are a subset of pulmonary nodules or masses with non-uniformity and less density than solid nodules. They are usually described as either non-solid or part-solid, and have a higher probability of malignancy.Cavitary: These nodules are ring shaped and identified by a cavity with darker intensity in the image. A pulmonary cavity is a gas filled area of the lung in the center of a nodule. The most commonly encountered cavitary nodules are malignant.

Subcategories:Isolated: nodules of this type are independent and do not appear connected to other anatomical structures.Juxtapleural: these nodules are attached to the neighboring pleural surfaces.Juxtavascular: this particular type of nodules is characterized by strong adherence to nearby blood vessels.

The number of cases for each category and subcategory is shown in Table 4.

### 3.3. Data Preprocessing

The raw data provided by a CT scanner need to be processed with the purpose of rendering it more meaningful and useful. The input of this preprocessing segment is a .dcm file representing the CT volume data of a scan, and the output is a set of appropriately sized images centered on lung nodules. Various methods and code were utilized from two separate solutions to the preprocessing part of the LUNA16 challenge [62,63].

The following steps were taken, as shown in Figure 10:1.Resampling: The slice images and parameters were extracted from the .dcm files; the slices had varying pixel spacing. It is more convenient to resample all scans from the database to the same pixel spacing, rendering the data more homogeneous. The selected values correspond to 0.5 × 0.5 (mm${}^{2}$/pixel); this spacing was chosen to avoid loss of details in the images, since it is lower than the minimum spacing of the whole dataset, thereby resulting in images with slightly higher resolution. The resampling was performed via third order spline interpolation.2.Lung segmentation: To remove all the undesired elements from the slice image and only keep the lung tissue, a segmentation method was used based on thresholding, connected component analysis and morphological operators. This helped reduce the problem space and is an important step for juxtapleural nodules. As the intensity values of both attached pleural and juxtapleural nodules are the same, traditional lung segmentation procedures result in concavity by exclusion of present juxtapleural nodules [20]. To counter this issue and to improve lungs’ area segmentation in cases of juxtapleural nodules a convex hull based method was employed. Afterwards the image is normalized to fit the range of −1000 HU to +400 HU; anything outside of this range is of no interest (bones with different radiodensity).3.Cropping: The images were cropped to a smaller and more manageable size. The final nodule images were centered and included enough surrounding tissue (length and width are double the nodule diameter). For the altered phantom dataset the nodule diameter corresponds to the actual diameter of the spheres, and the approximate centroid coordinates were annotated by the author. For the LIDC dataset the diameter and centroid were taken from the annotations provided for the LUNA16 Kaggle challenge. This work is mostly focused on the segmentation of lung nodules, which in a fully automated set-up should be preceded by a CADe system providing the coordinates of the nodule candidates. In a real setting where annotations are not available, the proposed method would be a block of a pipeline, located after detection and before feature extraction and classification.

## 4. Evaluation

There were two separate evaluation procedures for the two datasets:Objective: Carried out by using the altered phantom dataset, which was specifically generated for this study. Due to the simplicity of the phantom nodules, the images were altered to give more complexity. The results were evaluated by comparing ground truth with the segmentation mask using different metrics. This method was used to gauge how well the segmentation would perform in a CADx system, since it employed automated metrics as criteria.Subjective: Performed using data obtained from the public LIDC dataset and with the help of five subjects experienced in radiology. A questionnaire was compiled where each segmentation instance was shown and the subjects were asked to rate each instance with three scores.

This type of hybrid evaluation has certain advantages, by analyzing and comparing both results the segmentation algorithm can be evaluated in greater depth. In this fashion dissimilarities between the two methods and datasets can be examined, highlighting which features are prioritized when evaluating lung nodule segmentation accuracy. The objective method will be considered as the primary evaluation, and the subjective method will be used as comparison and validation for the efficacy of the altered phantom dataset.

### 4.1. Comparison Method

In order to establish how well the segmentation algorithm performs, another method is necessary for comparison; with this in mind, active contour models were chosen. Active contour models, also called snakes, are iterative image segmentation algorithms. Using active contour models, initial curves on an image need to be specified and then the curves evolve towards object boundaries. As input it requires the image to segment and binary mask specifying the initial contour, after the stated number of iterations the final contour is returned as a binary mask.

The active contour model used for segmentation is the Chan and Vese region-based energy model described in [14], the activecontour function uses the sparse field level set method, similar to the method described in [64], for implementing active contour evolution. In Figure 11 an instance of lung nodule segmentation with active contour models is shown.

The number of iterations has been set to 300 to make sure that the algorithm has enough time to accurately segment the nodule. The initial contour was set as a square whose sides were at a distance of five pixels from the image border. To regulate the degree of smoothness of the boundary, the parameter SmoothFactor was set to 0, to make sure that the irregularities on nodule borders were not unnecessarily smoothed. The tendency of the contour to grow outwards or shrink inwards was controlled by the parameter ContractionBias, which was set to 0, meaning that there was no preferred direction of evolution.

### 4.2. Objective

Both segmentation algorithms were applied on the nodules from the altered phantom dataset, and the resulting binary masks were compared to the ground truth. The performance evaluation was done via two metrics.

The first metric is the Dice similarity coefficient, which quantifies the overlapping area between the segmentation mask and ground truth. It is a spatial overlap index and a reproducibility validation metric often used for image segmentation [65]. For two sets A and B, it is defined as shown in Equation (Equation 3).
(3)$$Dice\left(A,B\right)=2\frac{\left|A\cap B\right|}{\left|A|+|B\right|}$$

Having two binary images, *A* and *B* become matrices containing ones and zeros. The intersection can be obtained as an element wise multiplication between the two matrices, and the denominator via a sum. To calculate the areas from the binary matrices, it is sufficient to calculate the sum of all elements in the matrices.

The second metric is the relative error of long-axis and short-axis diameter measurements. The measured lengths of the ground truth are compared to the measured lengths of the segmentation. Manual diameter measurements are the most widely used technique for quantifying the size of lung nodules in CT scans; this automated method will follow the same procedure. Following recommendations for measurement of nodules in CT images from the Fleischner Society [66], the long-axis diameter is taken as the maximal distance from two points on the nodule border and the short-axis diameter is taken as the maximum distance that is perpendicular to the long-axis. The values should be expressed to the nearest whole millimeter. These are usually performed manually by radiologists, for this automated procedure an algorithm was written that performs the same measurements. After calculating long-axis and short-axis diameters, what remains is to compare the measurements performed on the segmentation result with those performed on the ground truth. To this end, the relative error percentage is calculated for both axes, using the classical definition from Equation (Equation 4), where $m_{s}$ is the measurement obtained from the segmentation and $m_{gt}$ is the measurement obtained from the ground truth.
(4)$$\delta =|\frac{m_{s}-m_{gt}}{m_{gt}}|\cdot 100\%$$

### 4.3. Subjective

The LIDC database also contains annotations which were collected during a two-phase annotation process using four experienced radiologists. Each radiologist marked lesions they identified as non-nodule, nodule < 3 mm, and nodules >= 3 mm [67]. A ground truth can be estimated using these annotations, but trying to extract a ground truth by combining four different readings adds unnecessary complexity and can introduce errors [68]. For this study a more direct approach was chosen. Normally the radiologist would draw the ground truth of each nodule manually and then it would be compared using a metric, but manual annotation takes time and can be cumbersome, instead a different technique was devised based on mean opinion score (MOS). Which is a numerical measure of the human judged overall quality of an event or experience. In telecommunications, a MOS is a ranking of the quality of voice and video sessions. Most often judged on a scale of 1 (bad) to 5 (excellent), these scores are the average of a number of other human scored individual parameters. They can be utilized to evaluate the quality of almost any visual or audio media, it has been used to evaluate the quality of medical images [69] and also to evaluate image segmentation quality [70].

A four-part questionnaire (one for each category) was devised in order to evaluate the region-based fast marching method and for the preservation of diagnostic information. The questionnaire has not been validated before, as it was used for the first time in this study. It was filled out by five subjects experienced in radiology from Tianjin Chest Hospital. The questionnaire had the following structure:Introduction: A paragraph that explains the purpose of the document.Data: A paragraph the gives a overview on how the data was processed and how the nodules were segmented.Rating: In this section, the rating scale was established. The segmentation quality was rated on a scale from 1 to 5, where 1 is the worst possible score and 5 the highest. In previous studies where MOS was used to evaluate image segmentation [69,70], a single score was used. To obtain a more insightful evaluation, for each segmentation instance three different scores were assigned. Since this kind of evaluation has never been done before for lung nodule segmentation, the scores have been devised while keeping in mind which aspects are the most important. The scores are described as follows: -Area preservation: this score quantifies how close the area of the segmented nodule is compared to the ground truth.-Shape preservation: this score indicates how similar the shape of the boundary of the segmented nodule is compared to the ground truth.-Overall diagnostic quality: this score rates the general quality of the segmentation, while taking all the features of the nodule into account; it should quantify how well the segmentation captures all these features.To quantify these scores and to make the task clear, for each part examples are shown on how different segmentation instances can be rated. A short justification for the rating is given, followed by the scores and a figure displaying the segmentation instance evaluated, as shown in Figure 12a.Evaluation: in this final section each segmentation instance is displayed. Every nodule was segmented and shown twice, once using the fast marching method and once using active contour models. The instances shown in the questionnaire have been randomly shuffled to make it impossible to know beforehand which method was used. For each segmentation instance a figure is displayed with the results, with three editable text fields to input the scores, as shown in Figure 12b.

## 5. Results

### 5.1. Objective

The results of the objective evaluation performed on the altered phantom dataset are summarized in Table 5. Overall, the proposed method performs better than active contour models, having higher mean Dice score and lower mean diameter measurement errors. In addition, there is a noticeable difference in standard deviation, with the proposed method having lower values.

The results for solid-round nodules are shown in Figure 13. In this case FMM obtains superior scores for all metrics and demonstrates to work better than active contour models. FMM can handle these nodules well because they are solid with a convex shape and sufficiently regular border. FMM performs better than ACM in juxtapleural cases and all the segmentation instances with Dice score less than 0.5 correspond to shortcomings of the ACM method when dealing with juxtapleural nodules. Additionally, in juxtavascular nodules FMM manages to capture the border better, whereas ACM sometimes erroneously includes the vessels in the segmentation. Long-axis and short-axis diameter errors are in line with the Dice scores, due to the sufficiently regular border of this type of nodule, meaning that area overlap is consistent with diameter measures.

The results for solid-irregular nodules are shown in Figure 14. As in the previous case, FMM obtains superior scores for all metrics and overall works better than ACM. Similarly to round nodules, irregular nodules are solid and are well suited for this segmentation method. When it comes to juxtapleural nodules the behavior is similar as before, and also the juxtavascular cases are better handled by the FMM algorithm. Due to the irregular nature of the border, long-axis and short-axis diameter errors in some cases are not aligned with Dice scores.

The results for sub-solid nodules are shown in Figure 15. These types of nodules are hard to segment due to the non-homogeneous intensity of the nodule tissue and the presence of non-solid parts. Both methods struggle and obtain varied results, although FMM still obtains superior overall scores on all metrics. In most cases where FMM performs poorly, the result is under-segmentation; conversely, for ACM the problem is over-segmentation.

The results for cavitary nodules are shown in Figure 16. Cavitary nodules have proven to be the toughest targets for segmentation. Indeed, both methods perform poorly, even though FMM earns moderately higher scores on average. These types of nodules are arduous to segment due to the presence of the cavity. In case of FMM, if the nodule tissue around the cavity is too thin, it can cause problem since it tends to merge larger clusters. The ACM algorithm does not take the cavity into account and returns only a single contour and, in the worst case, it confuses the internal cavity boundary with the external nodule boundary. A more accurate segmentation of cavitary nodules with ACM could be possible by running the algorithm again on the segmented nodule, thereby segmenting the cavity and obtaining a second boundary. However, this approach would create new processing costs and difficulties.

In Figure 17 are some examples where FMM segmentation performed well and obtained high Dice scores, shown with their related Dice scores and the categories/subcategor-ies they belong to. In Figure 17a,d one of the advantages of FMM can be seen; i.e., it can properly handle juxtavascular cases as it excludes the vessels from the final contour. In Figure 17e,f the slight difference between ground truth and segmentation mask is attributed to FMM’s flaw in preserving detail with sharp and irregular borders. In Figure 17g,h are examples where FMM segments sub-solid nodules well despite their inhomogeneous intensity. Finally, Figure 17i shows an example where a cavitary nodule with a thick ring is segmented with a suitable internal and external contour.

In Figure 18 are some examples where FMM segmentation performed poorly and obtained low Dice scores, shown with their related Dice scores and the categories/subcategories they belong to. Most cases with low scores correspond to sub-solid and cavitary nodules, with the exception of Figure 18a where FMM is unable to accurately obtain the border due to the small size and irregular shape of the nodule. In Figure 18b,c sub-solid cases where the boundary is hard to distinguish are shown; thus, FMM does not manage to differentiate between nodule tissue and external tissue or anatomical structures. Figure 18d–f show examples where FMM cannot capture the thin nodule tissue around the cavity.

### 5.2. Subjective

The results of the subjective evaluation performed on the LIDC subset can be seen in Table 6. Altogether, active contour models perform slightly better than the proposed method, on all three scores. Furthermore, there is an apparent difference in standard deviation, with FMM presenting higher values.

Solid-round nodules are relatively easy to segment because of their simple shape and regular border. Both segmentation algorithms perform well and the average scores are close to 4. ACM has a slightly higher mean for all three scores. In most cases where FMM segmentation received a lower rating, it was penalized for not encompassing enough of the surrounding tissue, which the subjects deemed important to include. In Figure 19 the resulting scores are displayed for each nodule.

When it comes to solid-irregular cases, which have a more complex border and are more difficult to segment than the previous type, both segmentation algorithms obtain good results with high average scores. Similarly to the antecedent case, ACM has a slightly higher mean for all three scores. In these cases ACM gives a more detailed segmentation, because it better captures the sharp edges present in irregular nodules. On the other hand, FMM tends to give smoother and more convex shaped results. In Figure 20 the resulting scores are displayed for each nodule.

Due to the non-homogeneous intensity and part-solid components, sub-solid nodules can prove challenging when it comes to segmentation. Nonetheless, both segmentation algorithms received fair scores. In most cases where ACM was rated higher, it was mostly due to the algorithm including more of the non-solid part of the nodule in the segmentation. In Figure 21 the resulting scores are displayed for each nodule.

The presence of a cavity makes cavitary nodules particularly difficult to segment. Both algorithms obtained fair results, but in terms of average scores ACM was rated higher. This is due to FMM having difficulties with cases where the ring shaped tissue around the cavity is thin. In Figure 22 the resulting scores are displayed for each nodule.

MOS is calculated as the mean of the three subjective scores, in Figure 23 are some examples where FMM segmentation obtained a high MOS, shown with their related MOS and the categories/subcategories they belong to. In Figure 23a,d FMM segments the nodule accurately by leaving out the vessel from the final contour. Figure 23b,c show juxtapleural cases where the contour is segmented with sufficient detail even in case of irregular shapes. Shown in Figure 23e is a case where, despite the inhomogeneous intensity and indistinct border, FMM manages to provide an accurate result. Figure 23f shows a cavitary case for which FMM is well suited due to the thickness of the ring.

In Figure 24 are some examples where FMM segmentation obtained a low MOS, shown with their related MOS and the categories/subcategories they belong to. All the instances correspond to sub-solid and cavitary cases. In Figure 24a,b FMM segmented only the solid part leaving the rest of the nodule outside the contour. In Figure 24c,d the final contour erroneously excluded thin nodule tissue, and in Figure 24e a part of the nodule that is seemingly detached from the main body is ignored.

### 5.3. Comparison with Active Contour Models

Altogether, the fast marching method performed better with the objective evaluation on the altered phantom data, and worse with the subjective evaluation on the LIDC data. This phenomenon can be attributed to two factors: the datasets and the evaluation criteria.

The altered phantom dataset was constructed to be as close as possible to how real nodules appear on a CT image, replicating the most important features that different nodules possess. The only discernible difference in datasets can be attributed to the fact that ACM segmentation fails on most juxtapleural nodules from the altered phantom dataset, as opposed to the LIDC subset where this phenomenon does not manifest. Indeed, in the altered phantom dataset on average the removed pleura occupies a larger area in the image, causing ACM to ignore the nodule contour and only focus on the pleura boundary.

The other more impactful difference is caused by the evaluation criteria, that prioritise distinct aspects of segmentation. In the evaluation of solid-round nodules, the subjects gave higher ratings to juxtavascular nodules, which included parts of the blood vessels in the segmentation. Instead, the objective method focused more on rewarding cases where the structure of the nodule separate is maintained separate from any surrounding anatomical objects. When it comes to solid-irregular nodules, the subjects assigned higher scores to ACM in cases where FMM could not capture in detail the sharp edges along the nodule border. However, FMM performed better in case of large obstructions, which it can partly ignore, wherein ACM includes them in the segmentation. ACM has the upper hand in segmenting the boundary in a detailed way, and FMM excels in conserving the overall shape and area. In the case of sub-solid nodules, the fast marching method is more robust and works well in segmenting the solid part; for the non-solid part it is more conservative and sometimes it fails to capture it in its entirety, producing under-segmented results. The ACM segmentation is more open to adding the non-solid part, but it can fail and include too much surrounding tissue, generating over-segmented results. In the case of cavitary nodules, an important difference in evaluation is that the objective method takes the cavity into account when calculating the Dice score, but the subjects evaluated only based on the outside boundary. Generally FMM obtains better results when the part around the cavity is thicker.

### 5.4. Comparison with Deep Learning

An indirect comparison can also be made between FMM and recent segmentation techniques based on deep learning. For this reason, two of the works brought up in Section 1 were chosen. Cao et al. 2019 (DBResNet) [29] and Xiao et al. 2020 (3D-UNet) [32] both proposed lung nodule segmentation methods which obtained competitive results. Their proposals have been evaluated on the public LIDC dataset, and produced Dice scores of 82.74% and 95.30% respectively. The mean Dice scores for FMM segmentation results in all the categories belonging to the altered phantom dataset are shown in Table 7, along with the mean Dice scores obtained by 3D-UNet and DBResNet on the LIDC dataset.

Given that FMM was tested on a different dataset, it is impossible to tell with certainty which method performs better, but based on these scores it can be considered comparable to DBResNet and lesser than 3D-UNet. FMM performed worse than the DBResNet average on sub-solid and cavitary nodules, but in the LIDC dataset these types of nodules are much rarer than solid types; thus, they would have a small impact on the average Dice score achieved by DBResNet. The accuracy achieved by 3D-UNet is noticeably higher than FMM and will be considered a standard to strive for in future work, where improvements will be made to the algorithm to overcome its limitations and to reach a better overall accuracy.

### 5.5. Discussion

The region-based fast marching method performs excellently when it comes to solid nodules, where splitting into regions, clustering and merging results in an accurate segmentation. Regarding juxtapleural cases, the method works well since it takes the segmentation of the lungs’ area of the preprocessing stage into account. In juxtavascular cases, the method can effectively isolate the nodule from the surrounding vessels. In nodules that are usually tougher to segment, it gives good overall performance, with varied degrees of success in edge cases. Sub-solid nodules are handled quite well, except for extreme cases where the non-solid part is hard to differentiate from the adjacent tissue. In cavitary cases, the method gives an accurate segmentation in nodules with sufficiently thick nodule tissue surrounding the cavity, but struggles in some cases where this tissue covers a small area.

A direct comparison was performed with a classical image processing technique (ACM) and also an indirect comparison was made with two of the most recent deep learning techniques (DBResNet and 3D-UNet), both obtaining satisfactory results. The latter comparison was useful to set a new goal for the proposed method to aim for. The two different evaluations allowed to gain insight on how accurately the fast marching method can segment lung nodules, with this proposed method prevailing with the objective method and reaching comparable performance with the subjective method. Furthermore, the fact that both datasets produce similar outcomes with completely unrelated evaluation criteria implies that the nodules from the altered phantom dataset are similar to real nodules and successfully emulate their most relevant features. However, the developed method also has limitations and overcoming them will be the direction of continued future efforts.

Our method needs improvements on the preservation of features along the border, where some irregular contours are not accurately segmented. This behavior can be seen in Figure 17, Figure 18, Figure 23 and Figure 24 and also from the subjective evaluation where shape preservation scores tend to be lower than area preservation scores, seen in Table 6. Another aspect which needs to be improved upon is robustness for cavitary and sub-solid cases, seen in Figure 18 and Figure 24. In cavitary cases where FMM performed poorly, thin areas belonging to the nodule are not added to the segmentation mask. In part-solid cases where FMM faltered only the solid part was segmented, and in non-solid cases with poor performance the issue was over-segmentation.

In terms of advantages, the region-based approach is well suited to preserving the area of the segmented nodules, and with a future 3D extension could show improvements. Furthermore, the low computational complexity makes the process of segmentation quick and efficient. It can be concluded that a subjective method and an automated method, based on metrics such as Dice and diameter measurements, can provide complementary assessments. Both are beneficial for different reasons: the subjects can give a more detailed evaluations based on complex features that are of interest, wherein the objective metrics can gauge how well the segmentation would operate in a CADx system.

## 6. Conclusions

The segmentation method discussed in this paper takes a region-based approach to the fast marching method, which proved to be robust and can give an accurate segmentation of the nodule in most cases. It is able to properly handle juxtapleural and juxtavascular nodules, as no particular drop in performance was noticed for these types of nodules. The proposed technique was evaluated using an objective method applied on simulation data and a subjective method applied on real patient data: it achieved satisfactory results. Especially in the objective evaluation, it outperformed a classic segmentation technique (active contour models), provided comparable results to DBResNet, and resulted in lesser accuracy than 3D-UNet, leaving room for improvement. The proposed method demonstrated good overall performance and promise in computer-aided diagnosis applications. With further development, it could be used as part of a CADx system and provide fast and accurate segmentation results for a variety of nodule types. The obtained mask and measurement will be used to assess the clinical significance of the examined nodules.

In future work, the algorithm will be improved while we strive for state-of-the-art performance. To render the segmentation fully automated, an optimal parameter search will be implemented. An extension to 3D segmentation is planned as well.

## Figures and Tables

**Figure 1 sensors-21-01908-f001:**
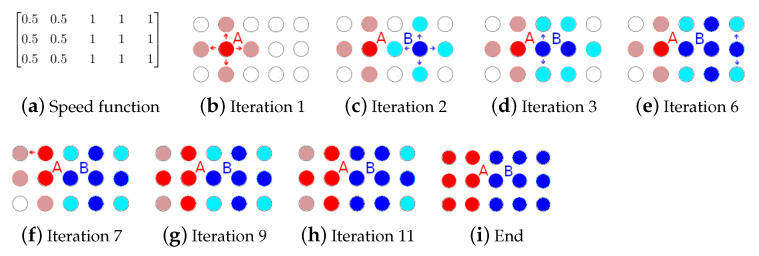
Examples of region assigning.

**Figure 2 sensors-21-01908-f002:**
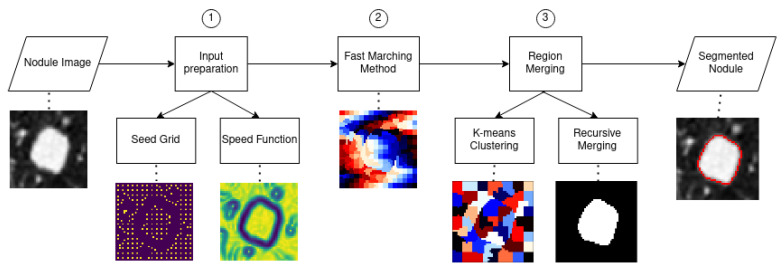
Flowchart of the proposed method.

**Figure 3 sensors-21-01908-f003:**
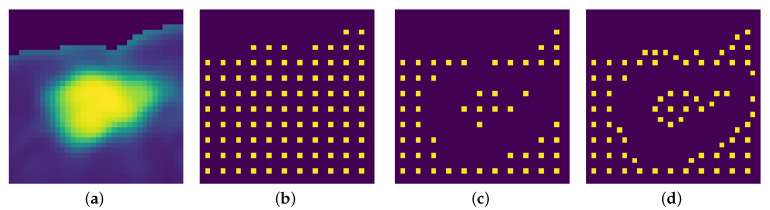
A seed grid generation example. (**a**) Input image. (**b**) Initial equidistant seed grid. (**c**) Seed grid after only deletion of points with high local gradient mean. (**d**) Seed grid after shifting and deletion.

**Figure 4 sensors-21-01908-f004:**
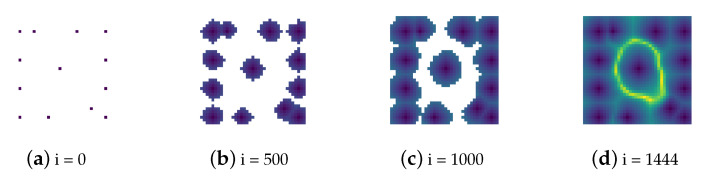
Evolution of the times matrix (*T*).

**Figure 5 sensors-21-01908-f005:**
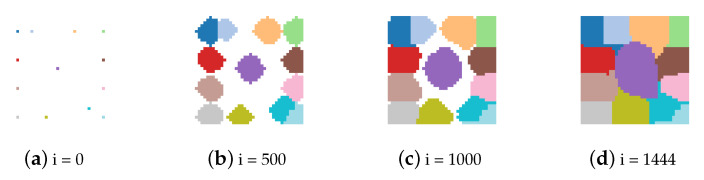
Evolution of the regions matrix (*R*).

**Figure 6 sensors-21-01908-f006:**
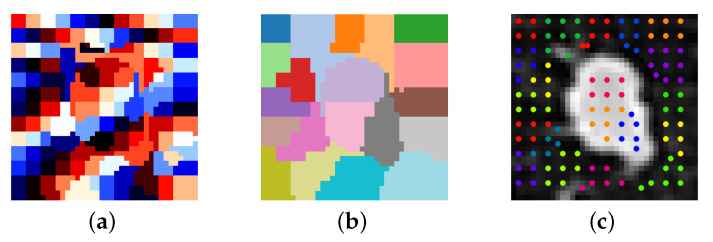
An example of grouping regions into clusters with k-means. (**a**) Regions. (**b**) Clusters. (**c**) Seed grouping shown over input image.

**Figure 7 sensors-21-01908-f007:**
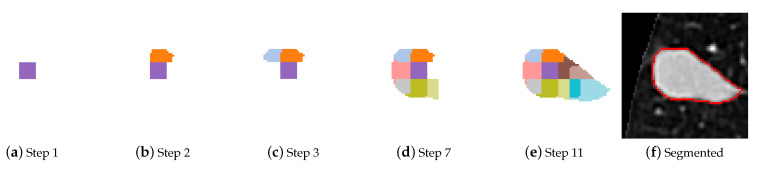
Merging of clusters, with a step counter.

**Figure 8 sensors-21-01908-f008:**
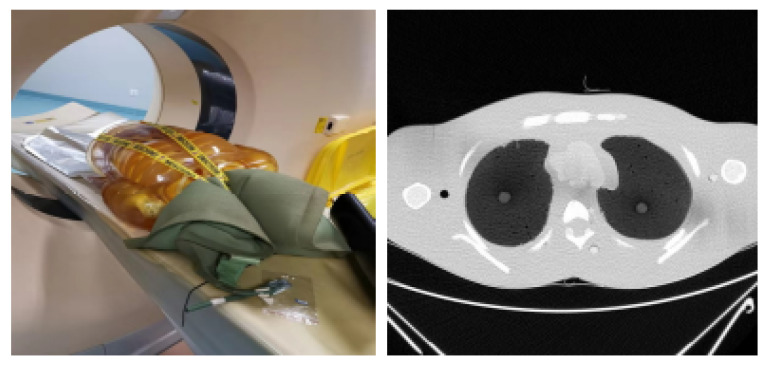
Lung phantom placed in CT scanner and a single axial slice.

**Figure 9 sensors-21-01908-f009:**
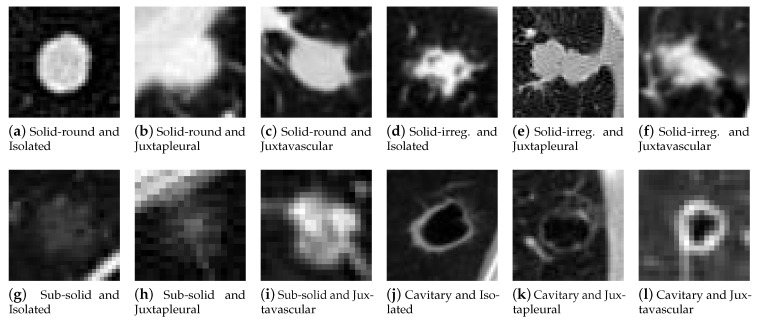
Examples of Lung Image Database Consortium (LIDC) nodules from every category and subcategory.

**Figure 10 sensors-21-01908-f010:**
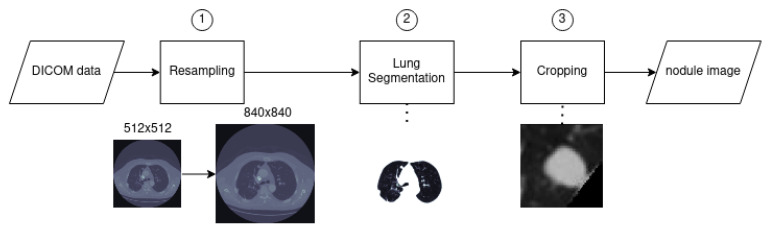
Preprocessing flowchart.

**Figure 11 sensors-21-01908-f011:**
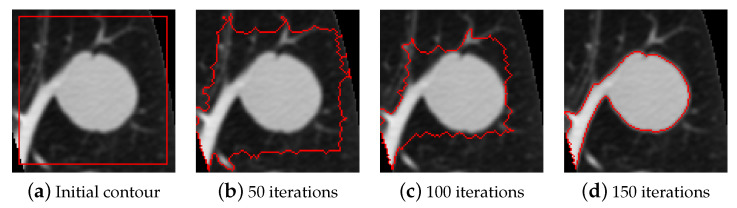
An example of activecontours segmentation.

**Figure 12 sensors-21-01908-f012:**
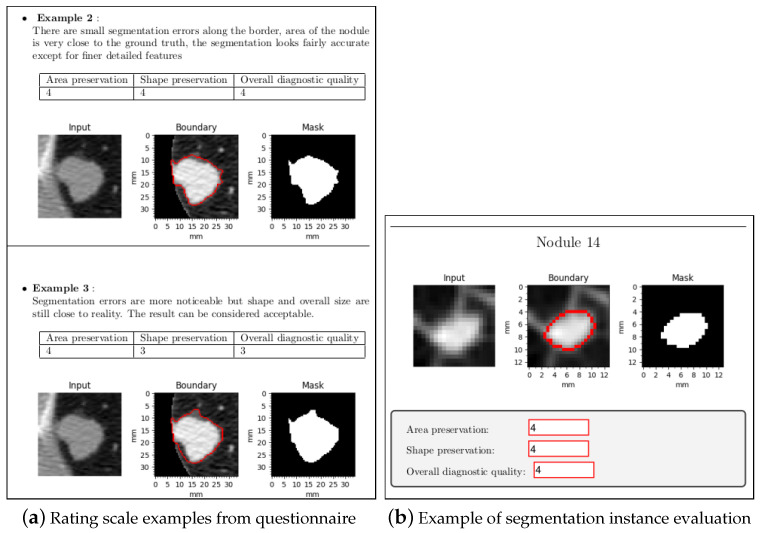
Excerpts from the questionnaire part one.

**Figure 13 sensors-21-01908-f013:**
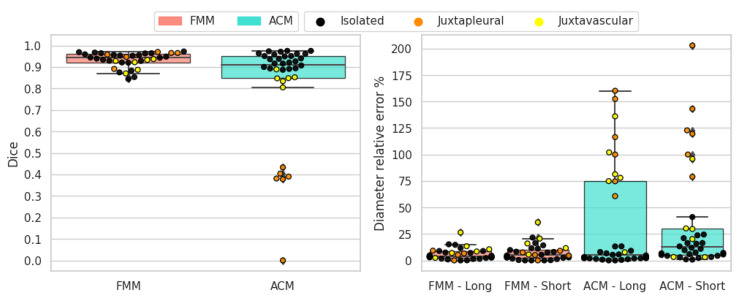
Solid-round nodules’ objective evaluation results as boxplots, overlayed with the values of individual cases divided into subcategories.

**Figure 14 sensors-21-01908-f014:**
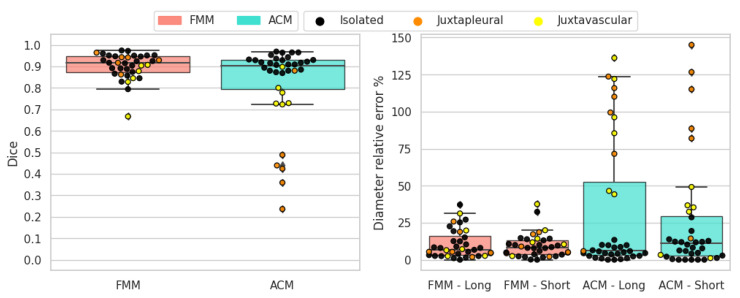
Solid-irregular nodules’ objective evaluation results as boxplots, overlayed with the values of individual cases divided into subcategories.

**Figure 15 sensors-21-01908-f015:**
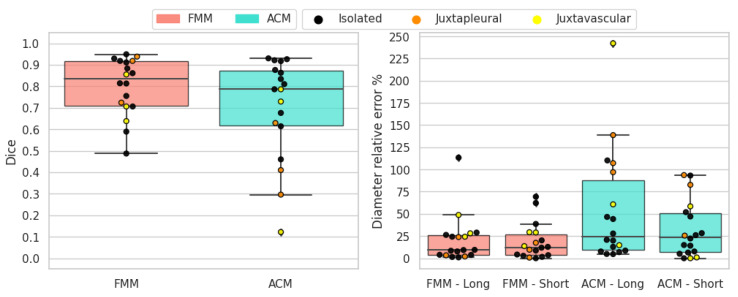
Sub-solid nodules’ objective evaluation results as boxplots, overlayed with the values of individual cases divided into subcategories.

**Figure 16 sensors-21-01908-f016:**
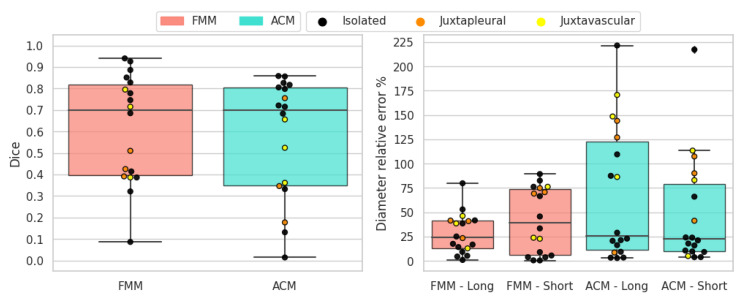
Cavitary nodules’ objective evaluation results as boxplots, overlayed with the values of individual cases divided into subcategories.

**Figure 17 sensors-21-01908-f017:**
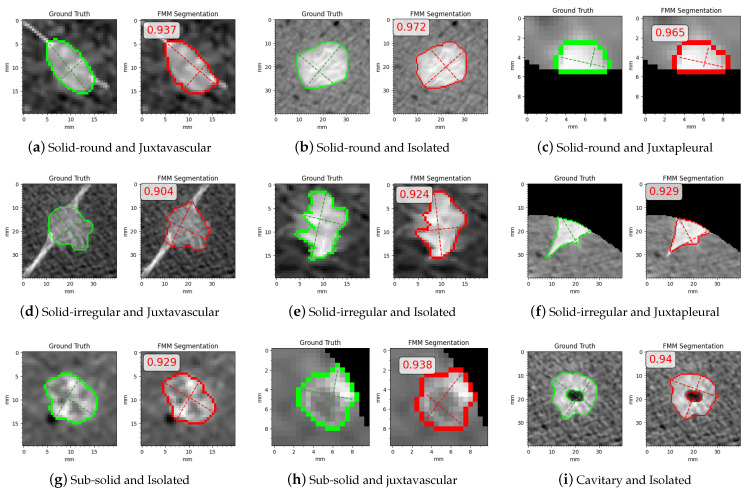
Examples with high Dice scores.

**Figure 18 sensors-21-01908-f018:**
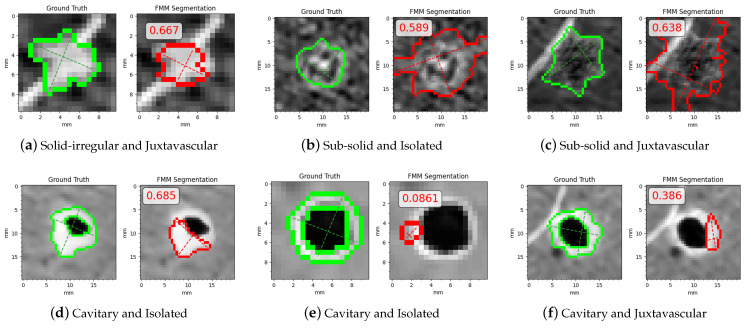
Examples with low Dice scores.

**Figure 19 sensors-21-01908-f019:**
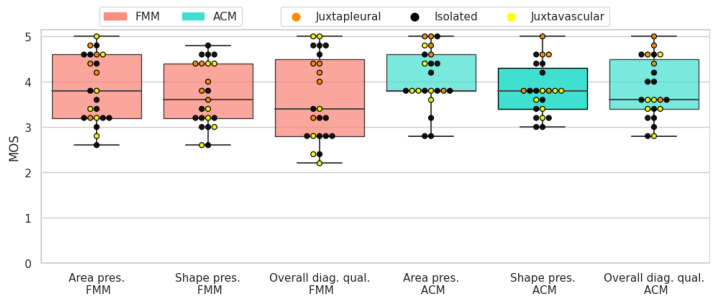
Solid-round nodules’ subjective evaluation results as boxplots, overlayed with the values of individual cases divided into subcategories.

**Figure 20 sensors-21-01908-f020:**
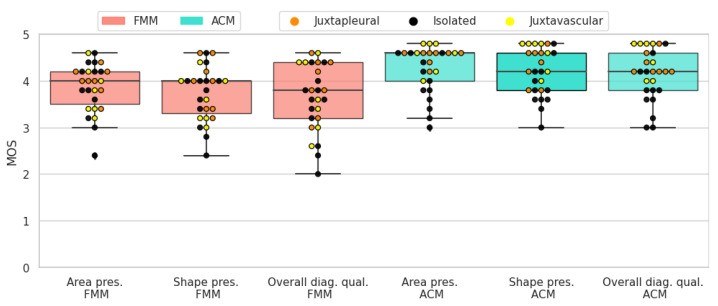
Solid-irregular nodules’ subjective evaluation results as boxplots, overlayed with the values of individual cases divided into subcategories.

**Figure 21 sensors-21-01908-f021:**
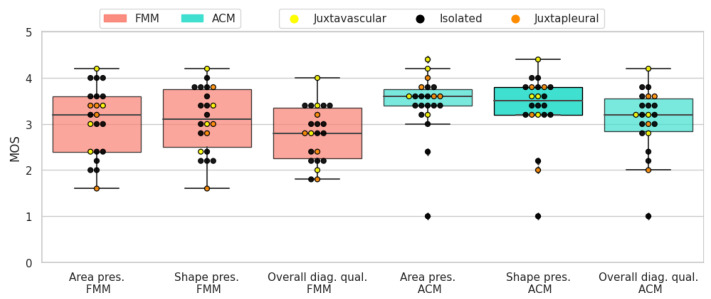
Sub-solid nodules’ subjective evaluation results as boxplots, overlayed with the values of individual cases divided into subcategories.

**Figure 22 sensors-21-01908-f022:**
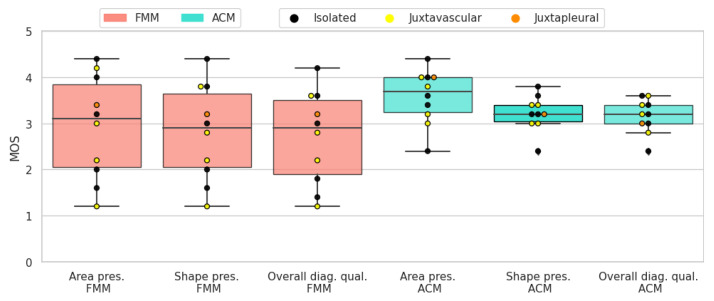
Cavitary nodules’ subjective evaluation results as boxplots, overlayed with the values of individual cases divided into subcategories.

**Figure 23 sensors-21-01908-f023:**
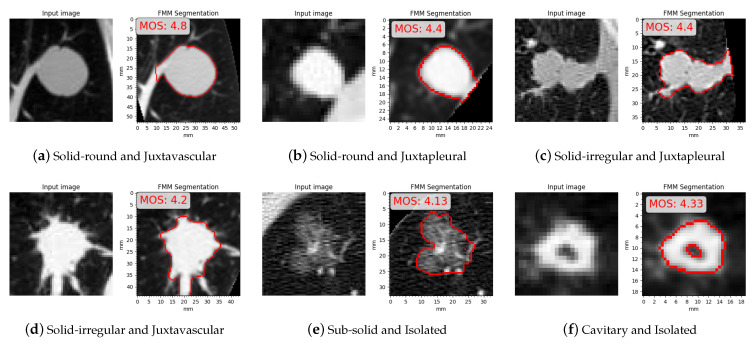
Examples with high mean opinion scores (MOS).

**Figure 24 sensors-21-01908-f024:**
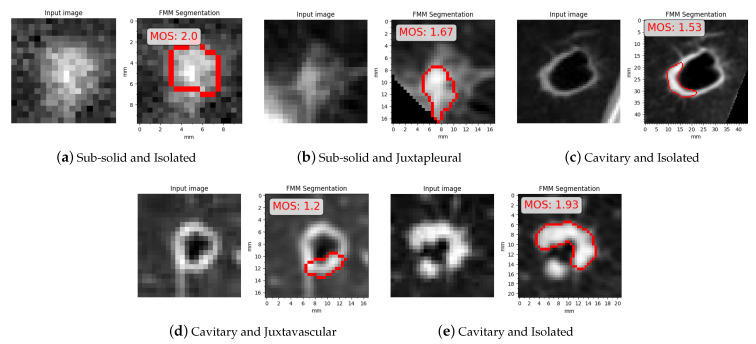
Examples with low mean opinion scores (MOS).

**Table 1 sensors-21-01908-t001:** A list of algorithms that are related to the proposed method.

Authors	Brief Description
Lung Nodule Segmentation	
Kostis et al. 2003 [10]	Applied iterative morphological filtering to remove vessels affixed to solid nodules
Kuhnigk et al. 2006 [11]	Employed morphological correction allowing to manage nodules regardless of size
Dehmeshki et al. 2008 [13]	Proposed a contrast based region growing method, that employs a fuzzy connectivity map
Chan and Vese 2001 [14]	Formulated segmentation as an energy minimization of an evolving contour seen as a level set
Farag et al. 2013 [15]	Used shape prior hypothesis along with level sets
Boykov and Kolmogorov et al. 2004 [16]	Framed the problem within a maximum flow optimization framework and used a graph cut method
Miao et al. 2016 [17]	GGO lung nodule segmentation with expectation–maximization algorithm
Miao et al. 2017 [18]	GGO lung nodule segmentation with ACM, solid and non-solid parts treated separately and combined
Li et al. 2020 [19]	Nodule segmentation with fuzzy C-means clustering and Gaussian mixture models
Wang et al. 2021 [21]	Enhanced total-variance pyramid and grab cut, boundary extraction with Gibbs energy functional
Lu et al. 2011 [22]	Proposed a stratified learning framework including supervised image segmentation
Hu et al. 2016 [23]	Utilized a Hessian-based vascular feature extraction procedure and classified nodules with a neural network
Gonçalves et al. 2016 [24]	Hessian-based strategies with a multiscale process that uses the central medialness adaptive principle
Jung et al. 2017 [25]	Separate solid and non-solid in GGO nodules using an asymmetric multi-phase deformable mode
Wang et al. 2017 [26]	MVCNN for nodule segmentation, which extracts features from axial, coronal and sagittal views
Ronneberger et al. 2015 [27]	U-Net architecture specialized for biomedical imaging
Wang et al. 2017 [28]	Central focused convolutional neural network for lung nodule segmentation
Cao et al. 2020 [29]	Dual-branch residual network for lung nodule segmentation
Qi et al. 2020 [30]	GGO nodules segmentation using CAD system based on CCN, analyzing growth and risk factors.
Funke et al. 2020 [31]	Trained a 3D-UNet model using using the STAPLE algorithm
Xiao et al. 2020 [32]	Combined the 3D-UNet and Res2Net architectures to create a new model
Hu et al. 2021 [33]	Hybrid attention mechanism and densely connected convolutional networks
Lung Segmentation	
Kavitha et al. 2019 [36]	Novel strip and marker-watershed based on PSO and fuzzy c-means clustering for lung segmentation
Kim et al. 2021 [37]	U-Net with self-attention for lung segmentation in chest X-rays.
Lung Nodule Detection and Segmentation	
Mekali et al. 2021 [20]	Lung boundary pixels and concave points extraction, separation of attached pleural from nodule
Huang et al. 2019 [38]	Detection with regional-CNN and segmentation with a FCN.
Other Pulmonary Disease Detection	
Polap et al. 2018 [39]	Diseased tissue detection via lung segmentation and bio-inspired algorithm
Ke et al. 2019 [40]	Detection of pulmonary disease with neuro-heuristic method
Santoso et al. 2020 [41]	ANFIS for detection of pneumonia and pulmonary tuberculosis
Ukaoha et al. 2020 [42]	ANFIS for diagnosis of COVID-19
Akram et al. 2021 [43]	COVID-19 diagnosis in X-rays via optimized genetic algorithm selector and naive Bayes classifier
Wang et al. 2020 [44]	Deep convolutional network for COVID-19 detection in X-rays

**Table 2 sensors-21-01908-t002:** Computed tomography (CT) scan parameters for both datasets.

	Phantom	LIDC
Tube current [mA]	30–197	40–582 (average 177)
Kilovoltage peak [kVp]	120	120, 130, 135, 140
Slice thickness [mm]	0.625	0.625–3.000 (average 1.7)
Pixel size [mm]	0.683	0.508–0.946 (average 0.66)

**Table 3 sensors-21-01908-t003:** The number of cases in each category and subcategory of the altered phantom dataset.

	Isolated	Juxtapleural	Juxtavascular	Total
Solid-round	24	6	6	36
Solid-irregular	24	6	6	36
Sub-solid	12	3	3	18
Cavitary	12	3	3	18

**Table 4 sensors-21-01908-t004:** The number of cases in each category and subcategory of the LIDC subset.

	Isolated	Juxtapleural	Juxtavascular	Total
Solid-round	12	5	6	23
Solid-irregular	12	8	7	27
Sub-solid	14	4	4	22
Cavitary	5	1	4	10

**Table 5 sensors-21-01908-t005:** Objective evaluation results—means and standard deviations of the criteria for each category, specifying the number of cases considered.

Category	№	Fast Marching Method			Active Contour Models		
		Dice	$e_{\mathrm{long}-\mathrm{axis}}$ [%]	$e_{\mathrm{short}-\mathrm{axis}}$ [%]	Dice	$e_{\mathrm{long}-\mathrm{axis}}$ [%]	$e_{\mathrm{short}-\mathrm{axis}}$ [%]
Solid-round	36	0.933 ± 0.034	5.94 ± 5.32	7.32 ± 7.48	0.819 ± 0.230	34.4 ± 49.2	33.6 ± 48.0
Solid-irregular	36	0.901 ± 0.059	10.6 ± 9.51	9.56 ± 8.09	0.821 ± 0.187	32.5 ± 44.9	25.2 ± 37.6
Sub-solid	18	0.799 ± 0.130	20.4 ± 26.0	18.6 ± 19.8	0.699 ± 0.230	54.2 ± 61.5	32.1 ± 30.9
Cavitary	18	0.614 ± 0.244	28.2 ± 20.0	41.8 ± 32.5	0.576 ± 0.269	68.4 ± 67.7	47.9 ± 54.7

**Table 6 sensors-21-01908-t006:** Subjective evaluation results as means and standard deviations of the scores for each category, specifying the number of cases considered.

Category	№	Fast Marching Method			Active Contour Models		
		Area pres.	Shape pres.	Overall d. q.	Area pres.	Shape pres.	Overall d. q.
Solid-round	23	3.84 ± 0.72	3.70 ± 0.68	3.62 ± 0.92	4.08 ± 0.63	3.82 ± 0.55	3.82 ± 0.64
Solid-irregular	27	3.86 ± 0.51	3.73 ± 0.57	3.65 ± 0.70	4.26 ± 0.49	4.19 ± 0.50	4.14 ± 0.53
Sub-solid	22	3.07 ± 0.72	3.09 ± 0.68	2.79 ± 0.58	3.42 ± 0.66	3.34 ± 0.73	3.08 ± 0.68
Cavitary	10	2.92 ± 1.06	2.80 ± 0.98	2.7 ± 0.96	3.57 ± 0.56	3.21 ± 0.36	3.15 ± 0.35

**Table 7 sensors-21-01908-t007:** Comparison between the fast marching method (FMM), DBResNet and 3D-UNet.

	Dice Score
Solid-round (FMM)	0.933 ± 0.034
Solid-irregular (FMM)	0.901 ± 0.059
Sub-solid (FMM)	0.799 ± 0.130
Cavitary (FMM)	0.614 ± 0.244
DBResNet	0.827 ± 0.102
3D-UNet	0.953 ± n/a

## Data Availability

The data presented in this study are openly available at github.com/marukosan93/fmm-lung-nodule-segmentation (accessed on 2 March 2021).

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
