# Peer review of "Lung Nodule Segmentation with a Region-Based Fast Marching Method"

_sensors, 2021, doi:10.3390/s21051908_

Round 1

Reviewer 1 Report

The paper proposes a segmentation algorithm based on the fast marching method for computer-aided diagnostics of lung diseases. The manuscript must be revised following the comments and suggestions presented below before it could be considered for acceptance.

Comments:

  1. Introduction section: extend the presentation of the research context. What are the main challenges in lung segmentation? What knowledge gap are you trying to bridge?
  2. Novelty is not clear. The advancement over existing methods seems small. Explicitly formulate the novelty of this paper at the end of the first section.
  3. Overview of the state-of-the-art related works is weak. Present a more comprehensive analysis and discussion on current methods for pneumonia and similar pulmonary disease identification from chest X-ray images. How about nature-inspired methods? Recently hybrid neuro-heuristic and neuro-fuzzy methods have demonstrated excellent results. Discuss these and other works for improving the overview: Doi:10.1007/s10044-020-00950-0,doi:10.1109/TII.2020.3022912, doi:10.3390/s21020369. Summarize the discussed works as a table.
  4. Provide a more detailed explanation on the limitations of the current methods for pneumonia and similar lung disease detection, i.e. how they deal with specific problems such as overfitting, class imbalance, low quality input images, etc.
  5. Add a structural schematics or block diagram of the proposed methodology.
  6. 160-L.166: use pseudocode to present the proposed algorithms.
  7. Explain parameter tuning in more detail. What parameter search / optimization methods do you use?
  8. Section 4.3: was the questionnaire you have used validated before?
  9. Present more informative figure captions, especially explain the meaning of subfigures in multi-part figures.
  10. Figures 12 -14 are not clear. What is an index? Is it image number in the dataset? How do you explain the large discrepancy for some images?
  11. Moreover, there are two separate Figures 12. Check figure numbering.
  12. Figures 15-18 have too many columns, hardly readable. Instead, summarize and use boxplots to compare different characteristics.
  13. Discuss the limitations of your method in the Discussion section.
  14. Conclusions section: use main experimental results to support your claims; add more in-depth insights into the possible impact of your research; discuss future work.

Author Response

We want to thank Rev.1 for their constructive comments and helpful suggestions,

1 - Following your suggestion we have extended the presentation of the research context. The main challenge of lung nodule segmentation is dealing with the heterogeneity of nodules on CT images and handling difficult cases like sub-solid, cavitary, juxtapleural and juxtavascular nodules. The gap we aim to bridge is the absence of an algorithm that radiologists can use to quickly obtain an accurate contour without the need for powerful computers and complex learning models. (Lines: 40-48 and 121-122)
2 - Thank you for your comment, as per your suggestion we have moved the novel contribution section at the end of the first section. Our purpose was to combine existing techniques which have low computational complexity to provide results on a par with those of other traditional methods in a simple and fast way. (Lines: 122-126)
3 - Thank you for pointing this out. We agree that the related works section needed improvement, and we extended it adding more current references, including doi:10.3390/s21020369 and doi:10.1109/TII.2020.3022912. However, our study had a different research purpose compared with doi:10.1007/s10044-020-00950-0 and discussion on current methods for pneumonia and similar pulmonary disease identification from chest X-ray images is not closely related to our work. (Lines: 80-87 and 106-117,Table 1)
4 - Thank you for this suggestion. It would have been interesting to explore this aspect. However, in the case of our study, it seems slightly out of scope because segmentation of lung nodules is not closely related with pneumonia and similar lung disease detection.
5 - We have added a flowchart of our proposed method. (Figure 2,Lines: 233-237)
6 - We agree that pseudocode is a better way of explaining that algorithm and have thus modified the manuscript to incorporate your suggestion. (Lines: 207-220)
7 - Parameter tuning has been done manually, and the setting we found proved to be adequate for the wide range of experimental data we used. However, we plan to devote future work to the goal of making it automatic and optimal. (Lines: 335-338)
8 - The questionnaire has not been validated before, as it was used for the first time in this study. (Lines: 527-528)
9 - We agree with this and have incorporated your suggestion throughout the manuscript.
10 - Following your suggestion in (12), we felt that using boxplots overlayed with the values of individual cases gives a much better visualisation of the results. The large discrepancies are due to cases which are tough to segment. With the improved figures, addition of subcategories and qualitative images they are easily reconductible to specific features. (Figures: 13-24)
11 - Thank you for pointing out this error, we have fixed the figure numbering.
12 - We thank you for the advice and have modified all figures containing the evaluation results into boxplots. (Figures: 13-16 and 19-22)
13 - As required, we discuss the limitations in the Discussion section. The method needs improvements on the preservation of features along the border and also in robustness for cavitary nodules. (Lines: 689-695)
14 - We agree with your comment and have modified the conclusions in accordance. (Lines: 709-718)

Reviewer 2 Report

The paper presents studies regarding the lung nodule segmentation based on region-based marching method.

There is presented the proposed segmentation method. The results are interesting, for the considered method the standard deviation having lower values. Also, there is conducted a comparison with deep learning method.

I positively appreciate the paragraph 6, that highlights the author’s contributions.

I recommend to place in an Appendix a short list of abbreviations that would contribute to the readability of the manuscript, even if the acronyms are well known in technical literature.

Minor English language improvements could be done.

Author Response

We are grateful to Rev. 2 for their comments.

We thank Rev. 2 for appreciating the author's contributions section, as we believe that it is important to clearly outline our contribution. Please notice that, upon request of the other reviewers we moved its contents to Sec. 1.

Thank you for your suggestion, we placed a list of abbreviations in the Appendix.

Thank you for pointing it out, we have made improvements on the language. 

Reviewer 3 Report

In this paper, the authors propose method for computer-aided diagnosis applications. This method has been compared to active contour models and to the Dual-branch residual network, reaching better overall accuracy than active contour models, and comparable results to that network. The experimental results show that the proposed technique can accurately segment nodules, with mean intersection over union 0.82-0.87 for solid cases, round and irregular nodules. The evaluation with two objective and subjective methods show that were applied on two different datasets, containing simulation data generated for this study and real patient data, respectively. The authors have proposed a segmentation algorithm based on the fast marching method, which can separate the image in regions with similar features, then merged by combining region growing with K-means. With the proposed methods and tools, the authors have a solution for lung cancer analysis and diagnosis by computed tomography data, for an accurate segmentation of lung nodules, which is of great importance.

I have some reviewer notes:

1.You can add and discuss some new literature sources, after 2017-2018.

2.Improve the conclusions on the base of the new data and findings.

3.Improve the conclusions with more comments about practical importance of your findings.

4.You can add information about some of the disadvantages of your findings and what will be your future work on this topic.

Author Response

We want to thank Rev. 3 for taking the time to assess your manuscript and for the constructive comments.

1. Following your suggestion, we have added new literature sources in section 1.2 containing recent successful traditional image processing methods and deep learning techniques. (Lines: 80-87 and 106-117)
2. Thank you for the suggestion, we have modified the statement made about comparison to deep learning, by also adding a more recent network to use for comparison. (Lines: 709-712)
3. We agree with your comment and revised the conclusion as to include practical importance of our findings. (Lines: 712-715)
4. Thank you for pointing this out, we have added information about the disadvantages and the direction of our future efforts.  (Lines: 689-695 and 716-718)

Reviewer 4 Report

The paper entitled “Lung Nodule Segmentation with Region-based Fast Marching Method”, describes a method for segmentation of lung nodules in tomography images using a Fast Marching Algorithm. The algorithm is validated using data from the LIDC dataset and from a phantom dataset attaining segmentation accuracies   expressed as intersection over union (IoU) of 0.877 and 0.823 for round and irregular nodules. Authors also incorporated a subjective validation.

This is a very important topic of research that gathers contributions from a large number of research groups [1] in recent years. Many of the results reported so far need improvements. The authors of the paper under review have reported a method for nodule segmentation that is interesting. However, the paper still needs a lot of work for improving the description and presentation of the results. Below I will include a set of comments aimed at improving the paper:

1) The introduction should be improved as the discussion concerning the related works discusses only a small set of references that is rather outdated.  As previously commented this is a very active field of research and there is an important set of contributions from 2019,2020 and 2021 that should be commented and used for comparison of the results. As an example see [1],[4] and [5].

2) Authors should improve the description of original contributions of the research reported in this paper. There are several contributions listed in section 6. However, the first contribution listed concerning the improvements in segmentation of juxtapleural nodules is not supported by actual results. The fact is that authors have used a rather non-standard classification of nodules reported in [2] that is rather outdated. They should better consider a more recent classification as the one reported in [1] and [3] that would enable quantification of yuxtapleural nodules and easy comparison with other recent methods. The last contribution reported in section 6 also requires improvements. In particular, the description of the objective validation is not well reported (see other comments below).

3) Authors claim that the proposed segmentation algorithm provides results comparable with respect to deep learning based techniques. Results are also close to active contour algorithms, so what are the advantages of using the proposed method? Authors should comment about this issue.

4) According to [1] the following processes are usually involved in Computer Aided Detection (CAD) of lung nodules:  image acquisition, segmentation of lung fields, detection of candidate nodules, and False Positive reduction. Authors present in Fig. 9 a diagram reporting the different stages of preprocessing. Authors should discuss how the nodules candidates are obtained as this stage should precede the cropping stage shown in figure 9.

5) As previously commented, the classification presented in [1][3] would allow to better compare this method with respect to other methods. Authors should consider adhere at this classification. Similarly, the Dice coefficient should also be calculated for comparing to other methods.

6) Authors claim they have used a subset of the LIDC dataset (section 3.2). They should report the number of cases used for validating the algorithm.

7) Authors should report examples of segmentation of nodules with the different shapes considered. The segmented shape should be presented along with the ground truth shape for cases where the IoU (or Dice) coefficient is high and cases where the IoU coefficient is low. This should be presented for both datasets.

8) Results presented in Table 2 should specify the number of images considered. Are the results for the phantom dataset or the LUNA dataset? Results should be presented separately of each of datasets.

9) In the comparison with active contours, authors should explain how the parameters of the algorithm were set. How the initialization of contour is obtained?

10) Authors should clearly state if the subjective test was performed using the phantom dataset or the LIDC dataset.

12) Section 5.3 adds confusion because authors only discuss results from the phantom, but previously they do not mention or reports any results about the phantom.

13) In Figure 3, the caption in each of the parts (a), (b) etc must be defined and explained. Similarly, for figure 4.

14) Concerning the cropping stage, it is not clear how this methodology could be used in a real application where annotations about the nodule candidates are not provided. Authors should discuss this issue or proposing a better method for centering the images.

15) Figures 12,13 and 14 includes an index that is not explained. Is the index the number of cases?

16) There are two Figure 12.

17) Description of Table 3 must report the number of cases considered.

[1] Shaukat, F., Raja, G., & Frangi, A. F. (2019). Computer-aided detection of lung nodules: a review. Journal of Medical Imaging, 6(2), 020901.

[2] Hansell, D. M., Bankier, A. A., MacMahon, H., McLoud, T. C., Muller, N. L., & Remy, J. (2008). Fleischner Society: glossary of terms for thoracic imaging. Radiology, 246(3), 697-722.

[3] MacMahon, H., Naidich, D. P., Goo, J. M., Lee, K. S., Leung, A. N., Mayo, J. R., ... & Bankier, A. A. (2017). Guidelines for management of incidental pulmonary nodules detected on CT images: from the Fleischner Society 2017. Radiology, 284(1), 228-243.

[4] Xiao, Z., Liu, B., Geng, L., Zhang, F., & Liu, Y. (2020). Segmentation of Lung Nodules Using Improved 3D-UNet Neural Network. Symmetry, 12(11), 1787.

[5] Funke, W., Veasey, B., Zurada, J., Frigui, H., & Amini, A. (2020, March). 3D U-Net for segmentation of pulmonary nodules in volumetric CT scans from multi-annotator truth estimation. In Medical Imaging 2020: Computer-Aided Diagnosis (Vol. 11314, p. 1131429). International Society for Optics and Photonics.

Author Response

We sincerely thank Rev.4 for their careful reading of our contribution, and for providing constructive criticisms and suggestions.

1) We agree with your suggestion and have thus updated our references to include recent contributions, thank you for suggesting [4],[5]. However, we thought it best not to include [1] as it is not strongly relevant to segmentation as it focuses on detection. (Lines: 80-87 and 106-117)

2) Thank you suggesting the more modern classification. We have modified our manuscript to adhere to it, while keeping the cavitary category and maintaining solid-irregular and solid-round nodules separate. In the results section (Figures: 13-16 and 19-22) it can be seen that there is no particular drop in performance for juxtapleural nodules. (Lines: 387-417 and 706-707, Figure 9,Table 4)

3) We have revised our comparison and added advantages in the discussion section. Our method is well suited to preserving the area of the segmented nodules and has low computational complexity. (Lines: 656-670 and 696-698,Table 7)

4) Our work is mostly focused on the segmentation of lung nodules and belongs in a Computer Aided Diagnosis (CADx) framework. Detection of nodules is not part of the aims of our proposed method. We have revised our article to make this point clearer. (Lines: 30-38)

5) As per your suggestion we have adhered to the more modern classification. We also changed our evaluation metric from IoU to Dice to allow easier comparison. (Lines: 488-496)

6) Thank you for the suggestion, we have reported the number of total cases and added a table detailing how many cases where split into each category/subcategory. (Line 384,Table 4)

7) We agree with this comment and have added examples for both datasets where dice is high/low and MOS is high/low. (Figures: 17-18 and 23-24)

8) Thank you for pointing this out. We modified the table to also report the number of images considered and also added two tables detailing the number of images for each category/subcategory. (Tables: 3-6)

9) We extended the description of active contours in section 4.1 to include information about the parameters set and the initial contour.  (Lines: 478-483)

10) We clarify that the objective test was performed using the phantom dataset, and that the subjective test was performed using the LIDC dataset. (Lines: 558 and 597)

12) In section 5.3 we discuss the results of both datasets, we modified our manuscript to state it more clearly. Results from the phantom are reported in section 5.1 as the objective evaluation was performed on the phantom data. (Lines: 627-628)

13) We thank for pointing out the lack of captions, we defined them and explained that the images are visualizations of the two matrices at different iterations. (Figures: 4-5)

14) The aim of our proposed method is to segment and measure lung nodules that have already been detected, in a real life application it would be preceded by a detector. Our work did not focus on detection but segmentation, we have revised the article to emphasize this point. (Lines: 445-449)

15) Having changed the figures into boxplots, which give a better visualization of the results, the index axis which was previously the case number is no longer present.

16) Thank you for pointing this out. We have fixed the issue.

17) As per your request we have added an additional column containing the number of cases considered for each category. (Tables: 5 and 6)

Round 2

Reviewer 1 Report

The paper has been improved somewhat with minor changes introduced by the authors. But my main concerns regarding the novelty of this study and its contribution over previous works remain. (Fast marching is a 25-year old method). The work is too much incremental. There are many similar studies published now. But the authors fail to formulate the main problems faced by lung segmentation and fail to formulate the contribution of this study adequately. The related works section remains chaotic, poorly organized and unstructured. The review should be addressed more systematically, by discussing separately different related subproblems and classes of research solutions such as fuzzy methods, heuristic methods, machine learning, deep learning and hybrid methods . The analysis does not provide insight into the limitations of methods and why yet another new method is needed.  There are many figures in the results section of the paper, but they are not sufficiently discussed in the text of the paper, so it is not clear what these figures are trying to show or prove.

I repeat my other comments as these there ignored by the authors in the revision:

  1. Provide a more detailed explanation on the limitations of the current methods for pneumonia and similar lung disease detection, i.e. how they deal with specific problems such as overfitting, class imbalance, low quality input images, etc.
  2. Explain parameter tuning in more detail. What parameter search / optimization methods do you use?
  3. The questionnaire has not been validated before, so its results are not reliable. Moreover, only five respondees provided answers, which makes the results not statistically meaningful.
  4. Discussion on the limitations in the Discussion section should be more specific. For example, in L. 691, what distance is noticeable?

Author Response

We sincerely thank you for your further effort and your helpful suggestions aimed at improving our manuscript. We did not mean to ignore your previous comments, we rather thought we had addressed them.

We have revised and reorganised the Related work section by expanding the research context and categorization. We added information about limitations of some lung nodule segmentation methods which our work is trying to overcome. (Lines: 55-172 and 179-186, Table 1)

We agree about your suggestion regarding lack of discussion for some of the figures in the Results section, and have modified the manuscript accordingly. (Lines: 656-662 and 664-669 and 697-701 and 703-706)

1 - Thank you for your comment. We do not claim to propose a systematic review paper; we address and try to motivate a method for image analysis, which we hope will find applications in the medical imaging field. Thus, we believe that a more detailed analysis of problems such as pneumonia and similar lung disease detection are out of the scope of our research. 

2 - We appreciate the suggestion and have added a more detailed explanation of how the parameters were chosen. (Lines: 395-401)

3 - We agree that the questionnaire could be developed further. However, it has been conceived by a professional practitioner (author Y.M.) who deemed it adequate for a preliminary test. At this stage of the research, and given the workload in this specific clinical sector due to this pandemic, it did not seem advisable to engage a larger group of researchers to improve it. Furthermore, we believe that a more rigorous statistical analysis would be out of the scope for this paper, as it would be more appropriate for a medical paper. 

4 - Thank you for pointing this out, we have discussed the limitations in a more specific manner including references to Figures/Tables where the mentioned phenomena can be seen. (Lines: 772-779)

Reviewer 4 Report

The paper entitled “Lung Nodule Segmentation with Region-based Fast Marching Method”, describes a method for segmentation of lung nodules in tomography images using a Fast Marching Algorithm. The algorithm is validated using data from the LIDC dataset and from a phantom dataset attaining segmentation accuracies   expressed as mean Dice score of 0.933 and 0.901 for round and irregular nodules and 0.799 and 0.614 mean Dice score non-solid and cavitary nodules

In this version of the papers, authors have improved the paper by incorporating the corrections and suggestions indicated by the reviewers. In my opinion the paper represents now an important contribution to the segmentation of lung nodules in computed tomography. The paper could be accepted for publication.

Author Response

We would like to thank Rev.4 for their valuable suggestions and for having allowed us to improve our paper.

Round 3

Reviewer 1 Report

All my concerns were addressed by the authors and the paper has been properly revised. I recommend the paper to be accepted.